# The Role of Extended *CFTR* Gene Sequencing in Newborn Screening for Cystic Fibrosis

**DOI:** 10.3390/ijns6010023

**Published:** 2020-03-21

**Authors:** Anne Bergougnoux, Maureen Lopez, Emmanuelle Girodon

**Affiliations:** 1Molecular Genetics Laboratory, CHU Montpellier, EA7402 University of Montpellier, 34093 Montpellier CEDEX 5, France; anne.bergougnoux@inserm.fr; 2Molecular Genetics Laboratory, Cochin Hospital, APHP. Centre, University of Paris, 75014 Paris, France; maureen.lopez@aphp.fr

**Keywords:** cystic fibrosis, newborn screening, DNA analysis, next generation sequencing, extended genetic analysis

## Abstract

There has been considerable progress in the implementation of newborn screening (NBS) programs for cystic fibrosis (CF), with DNA analysis being part of an increasing number of strategies. Thanks to advances in genomic sequencing technologies, *CFTR*-extended genetic analysis (EGA) by sequencing its coding regions has become affordable and has already been included as part of a limited number of core NBS programs, to the benefit of admixed populations. Based on results analysis of existing programs, the values and challenges of EGA are reviewed in the perspective of its implementation on a larger scale. Sensitivity would be increased at best by using EGA as a second tier, but this could be at the expense of positive predictive value, which improves, however, if EGA is applied after testing a variant panel. The increased detection of babies with an inconclusive diagnosis has proved to be a major drawback in programs using EGA. The lack of knowledge on pathogenicity and penetrance associated with numerous variants hinders the introduction of EGA as a second tier, but EGA with filtering for all known CF variants with full penetrance could be a solution. The issue of incomplete knowledge is a real challenge in terms of the implemention of NBS extended to many genetic diseases.

## 1. Introduction

For more than 40 years, newborn screening (NBS) programs for cystic fibrosis (CF) have been implemented across the world in Caucasian populations as pilot, regional, or national programs [1,2,3,4,5,6]. While early *CFTR* gene analysis is a key tool for the diagnosis of CF, for adapted follow-up, mutation-guided therapy, and genetic counseling, its place and extent in a core NBS program has long been a matter of debate. At present, there are a variety of established NBS programs [1], all starting with immunoreactive trypsinogen (IRT) measurement. DNA analysis is part of the majority of programs, most often performed in the second tier on the same dried blood spot as IRT measurement, and principally with search for a limited number of frequent variants. Some programs include extended *CFTR* gene analysis with next generation sequencing (NGS) focused on coding regions (referred to as extended genetic analysis (EGA)), mainly because of heterogeneous populations where the distribution and the frequency of CF-causing variants vary. These programs, however, lead to the detection of a high number of inconclusive cases, also referred to as Cystic Fibrosis Screen Positive, Inconclusive Diagnosis (CFSPID) in Europe or CFTR-related metabolic syndrome (CRMS) in the US, compared to the number of CF cases [1,7]. The definition of CRMS/CFSPID includes infants with a sweat chloride value between 30–59 mmol/L and zero or one CF-causing variant, or a sweat chloride value below 30 mmol/L and two *CFTR* variants, at least one of which has unclear phenotypic consequences [8,9]. Beyond technical and medical aspects, the choice of a NBS strategy is driven by the mutation spectrum in the screened population, the laboratory facilities, the health care system, the legal and economic aspects, and the acceptability by the population [10,11,12,13]. There is thus no universal model of NBS strategy. Nevertheless, there are minimal recommendations which have been issued by the European Society of Cystic Fibrosis (ECFS) [14], including, as performance criteria, a minimal positive predictive value (PPV) of 30% and a minimal diagnostic sensitivity of 95%. It is also an objective to minimize as much as possible the number of inconclusive cases and carrier detection. In the present article, we review the variety of NBS programs set up that include DNA analysis, along with their performance. The value, challenges, and drawbacks of *CFTR* EGA are then discussed, and then future prospects are eventually considered—in particular, in the view of a shift toward an extended NBS of a set of hereditary diseases by wide genomic analysis.

## 2. Overview of Newborn Screening Programs

A review of the literature was conducted to identify current NBS programs for CF and to update performance data, notably from the important review across Europe [1]. It is, however, possible that strategies have changed since the last available data. Where data were old and not detailed, they were not included in Table 1. In Europe, 22 countries have implemented NBS programs, 20 as national and two as regional programs. Fifteen programs use DNA analysis, mostly as a second tier after IRT measurement. Twelve countries use a variant panel only, and three have implemented EGA (Figure 1a). In North America, all Canada provinces follow an IRT/DNA panel strategy, as do the great majority of US states [5,6] (Figure 1b). Only six US states do not use DNA analysis in their core program. The states of California, Hawaii, and, recently, New York, which are composed of multi-ethnic and/or non-white populations, have introduced EGA as a third tier after analysis of a variant panel [5,7] (Personal Communication [15]). In Central and South America, some programs have been implemented, e.g., since 2001 in Brazil [16], but few have introduced DNA analysis in their strategy [17] (Figure 1c). In Australia and New Zealand, where NBS was implemented in 1989 and 1986, respectively, DNA analysis using mutation panels is now part of the strategy in the second tier [18,19] (Figure 1d).

Numerous programs have included a safety net protocol with the aim to identify CF infants who would be missed by testing variant panels because they carry two rare variants. Depending on the program, neonates with no variant but ultra-high IRT undergo either a second IRT measurement at 14–21 days or a direct sweat testing. In Denmark and the Netherlands, the safety net protocol leads to performing EGA. Safety net protocols lead to limiting false negatives cases, hence increasing the sensitivity of NBS programs, which is particularly effective in ethnic minority communities.

Comparing performances between different strategies using DNA analysis is a difficult task because of the considerable variability of protocols (Table 1), the changes over time in a country, the number of years since implementation, the duration of follow-up, the strategies and sensitivity of DNA analysis, and the monitoring of performance. Even when equivalent strategies are applied, there is variability in each step of the protocol, such as in the choice of the IRT cut-off value, the number of frequent variants screened by the panel, the use of a third or even fourth tier to minimize the number of children referred for sweat testing, or the safety net protocol (Table 1).

It is recommended to screen all pathogenic variants found at an incidence of 0.5% or higher according to geographical origins, or all variants accounting for more than 80% of CF alleles in the screened population [20]. The number of tested variants in NBS programs varies from 1 (F508del in Denmark and some states of the US [5,21]) to 388 (in New York State) (Personal Communication [15]). 

### 2.1. Programs Including Variant Panels Only

Most of the NBS protocols use a panel of frequent CF-causing variants after the first raised IRT value. Only Serbia, Ukraine, and some US states have included a second IRT measurement on a new blood sample before testing the panel, and Germany has included pancreatitis-associated peptide (PAP) measurement as a second tier before DNA analysis. The UK and Norway use a second DNA panel in neonates found to carry a variant of the first panel. 

In most of these programs testing for a variant panel, infants carrying only one identified variant are referred for sweat testing in order to distinguish affected infants from healthy carriers. In the UK and Luxembourg, a second IRT measurement on a new sample is performed to select neonates found to carry one variant who will undergo sweat testing. 

Inclusion into NBS programs of DNA analysis using variant panels appears to be at the benefice of both sensitivity and PPV compared to IRT/IRT protocols [1,11], with fewer false positive infants referred for sweat testing. However, despite the use of panels adapted to each population to achieve an optimal mutation detection rate, there remains disparity regarding sensitivity of the programs, with the majority above 95% (10/12 countries) according to ECFS recommendations [14]. Moreover, PPV varies from 3% to 76%. Of the 14 available PPV data sets, only four are above the ECFS target of 30%. Norway and the UK use DNA analysis in two successive panels, and the UK has also included a second IRT measurement after DNA analysis to select infants who will undergo sweat testing, which has raised the PPV up to 76%. 

The CF:CFSPID ratio also varies between countries that use a DNA panel strategy, from 17:1 in Switzerland to 1:1 in Norway, along with the carrier frequency, from 1/9.5 in Wisconsin to 1/44 in Germany. This could be explained by different mutation detection rates of variant panels, implementation of a safety net protocol, IRT cut-off, and the frequency of carriers in each population. Indeed, the number of healthy carriers detected increases with the mutation detection rate [11].

### 2.2. Programs Including EGA

Six programs have implemented EGA in their strategy, applied after a first step of DNA analysis when one CF-causing variant is found. Denmark tests a single variant [21], the Netherlands a panel of 35 [37], California State tests a panel of 40 [7], Hawaii State a panel of 97 [5] and New York State a panel of 388 (Personal Communication [15]). Poland applies *CFTR* sequencing in two tiers: analysis of *CFTR* regions covering 77% of Polish variants, then all *CFTR* coding regions (Personal communication [39]). There is no available data on the performance of the programs in Hawaii State and in New York State where the strategy recently changed. While PPV in the Polish program is 26%, it is above 30% in the others, varying from 34% to 85% (Table 1). By contrast, the sensitivity displayed in the Polish program is 100%, compared with 90–92% for the other three.

Denmark have chosen a low IRT cut-off (50 ng/mL) and test for F508del which is carried by approximatively 96% of the CF patients on at least one allele [21]. F508del would be carried by only 79% of the CF patients in Poland on at least one allele, according to the 54.5% F508del allelic frequency among CF alleles [40]. The high frequency of F508del allele in Danish CF patients may also explain the better CF:CFSPID ratio in Denmark (7:1), compared with the ratio in California (0.65:1). The CF:CFSPID ratio of 4:1 in the Netherlands is probably due to the additional step of PAP measurement that minimizes the number of positive screened newborns. 

The rate of CF carriers detected through NBS also varies, from 1/15 in Poland to 1/28 in the Netherlands.

## 3. Values and Challenges of *CFTR* Extended Genetic Analysis 

Assessing the value of EGA in NBS strategies is not a simple matter, given the few experiences documented, the different protocols followed, and the variable number of years since implementation. The analysis and the discussion below take into account these experiences, as well as considerations beyond to assess the value and challenges of EGA, either as a third-tier step after testing for a variant panel or directly as a second-tier step. They are summarized in Table 2.

### 3.1. Values of CFTR Extended Genetic Analysis in Newborn Screening Programs for CF

#### 3.1.1. Reduction of NBS False-Positive Results and Improvement of the PPV

Inclusion of EGA as a third tier after testing a variant panel appears to reduce the number of NBS false-positive results, thus the consecutive number of unnecessary sweat tests, and to improve the PPV as compared to programs that include testing for variant panels only (Table 1) [41]. Sweat chloride measurement, a key component for confirmation of CF, indeed requires a specific local CF center organization with a dedicated laboratory, as well as strong experience and routine practice of physicians for optimal performance and high quality [42,43]. 

#### 3.1.2. Best Equity of CF Screening between Populations 

One of the main rationales and benefits of implementation of EGA in NBS programs for CF is better equity for multi-ethnic communities compared to variant panels [43]. Even if a unique panel has been initially designed to cover the most frequent CF variants of the local population, migration flows modifying the ethnic mix would require recurrent updates to maintain high sensitivity of the test. Any DNA panel, even of large size, will miss rare variants. 

#### 3.1.3. Increased Knowledge of the Phenotypical Spectrum of CFTR Variants 

EGA also opens the opportunity to enrich our knowledge on the phenotypical spectrum and penetrance associated with *CFTR* variants, especially variants of varying clinical consequences (VVCC) and variants of unknown significance (VUS). Long term follow-up of patients who carry a VUS in *trans* of a CF-causing variant would help define variant pathogenicity and enrich locus-specific databases (CFMDB [44], CFTR2, [45], and *CFTR*-France, [46]). 

#### 3.1.4. Earlier Diagnosis and Access to Treatment

In the era of precision medicine, extended sequencing of the *CFTR* gene in IRT-positive neonates would detect all actionable variants for which efficient modulators of CFTR protein are now available. An early molecular diagnosis for CF would thus ensure early access to treatment. Comparison of the delay at the first medical visit may not seem better in programs that include EGA, but the delay until the two CF-causing variants are detected would be much shorter. 

### 3.2. Challenges

#### 3.2.1. Healthcare System Organization

Implementation of EGA in NBS programs for CF raises major challenges. Not only should the costs of the core strategy, which may vary considerably [7,13,47], be taken into consideration but so should those incurred by sweat testing if EGA is performed in the second tier, variant functional testing, genetic investigations in the parents, medical visits, and long-term follow-up of all screen-positive infants, either diagnosed with CF or with an inconclusive diagnosis. 

#### 3.2.2. Technical Issues

##### NBS Sequencing Platforms Development or Reorganization

Sequencing for NBS can be done on the same platform as for routine diagnosis already in place in laboratories. However, NBS requires a specific organization with optimized preanalytical protocols on dried-drop specimen and a specialized molecular genetics team for *CFTR* variant interpretation. Moreover, priority has to be given to the NBS process in order to respect ECFS recommendations about the timeliness of results. 

Interestingly, the molecular strategy applied in the states of California and New York includes a single technical assay for DNA analysis, that is, NGS sequencing of all *CFTR* exons and recurrent deep intronic variants, with a two-step bioinformatic analysis: an initial, predefined variant panel is run simultaneously to a deep scan that is masked, and the mask is lifted if only one CF-causing variant is detected. This unique molecular strategy may further improve the timeliness of results as compared to other approaches.

##### Sequencing and Bioinformatics Limitations for Variant Detection

Technical limitations include sequencing errors in homopolymer-rich sequences or in pseudo-homologous genomic regions [48] and the risk of missing short insertions or deletions because of misalignment of sequence reads. It is thus recommended to use several variant callers in the NGS pipeline in order to maximize variant detection efficiency [49], and this should be considered in the evaluation of costs. Inappropriate or insufficient design coverage could lead to missing large deletions or insertions and variants located in deep intronic regions. The design should thus be adapted and continuously updated accordingly [50,51]. Low DNA quality and quantity extracted of dried-blot samples are other constraints for the detection of large deletions and insertions [52].

Positive results should be confirmed by an independent assay on a second DNA dilution for identity monitoring, as in other programs. Using EGA, a higher number of variants is expected to require confirmation because of an increased detection and the possible technical limitation of NGS. Familial segregation analysis is also essential to confirm compound heterozygosity and to avoid false-positive results due to numerous complex alleles reported in the *CFTR* gene [46,53,54]. Parents’ blood sampling for DNA analysis should therefore be carried out optimally during the first visit so that medical care is not delayed. These essential experiments, whose numbers will inevitably increase with EGA being included in NBS, especially as a second-tier step, have to be considered for medico-economic studies.

#### 3.2.3. Byproducts of NBS for CF and Ethical Issues 

##### Increased Detection of Inconclusive Diagnosis

The identification of a substantial number of VUS or VVCC in neonates without any clinical symptoms and leading to an inconclusive diagnosis is a major drawback observed in the few experiences of EGA in NBS. In strategies that include EGA as a core component, neonates found to carry two variants (one of which being a VUS) and to have a negative sweat test fall into the category of inconclusive diagnosis, while in other strategies that include variant panels only, these neonates would be classified as healthy carriers. Indeed, since the sweat test is negative, DNA analysis stops at the step of variant panel and the VUS will not be identified. Implementation of EGA as a second-tier analysis after a raised IRT would lead to the further detection of infants with an inconclusive diagnosis who would have a negative sweat test and carry other kinds of genotypes, composed of two VVCC, a VVCC and a VUS, or two VUS. 

Despite accumulation of considerable data on *CFTR* variants in locus-specific variant databases (CFMDB, CFTR2, *CFTR*-France), epidemiological data and genotype–phenotype correlations still miss most rare genetic *CFTR* variants, which are sometimes restricted to one family. Furthermore, the number of new variants is steadily increasing, as illustrated in the California program, where 78 out of 303 rare variants were novel [7]. On the other hand, databases may provide discordant information that one should be aware of, notably because of their different designs. Data in CFTR2 are provided from national registries of CF patients and thus represent the “tip of the iceberg” of all possible phenotypes associated with a given genotype. By contrast, *CFTR*-France collects genetic and clinical data from patients with CF and CFTR-RD and from asymptomatic individuals, and so some variants are differently classified in the two databases. In addition, variants that are not referred to in the list of 432 described in CFTR2 have been classified as VUS in several programs [7,55,56], despite their description in the literature and other databases, either as CF-causing or as non-CF-causing. As a consequence, the number of CRMS/CFSPID observed in programs that include EGA may be overestimated and the ratio of CF:CFSPID artificially biased.

The risk that neonates with an inconclusive diagnosis will develop symptoms consistent with CF, even in a less typical form, with a possible delayed positive conversion of the sweat test [57,58] is still unknown. Nevertheless, parents should be informed on the longer-term risks that their child may develop clinical symptoms of CF. Ex vivo electrophysiological explorations on nasal epithelium, organoids, or rectal biopsies and in vitro experiments on heterologous immortalized cell systems reproducing DNA changes may help interpret variant pathogenicity and better evaluate the risk of developing clinical CF. A follow-up longer than two years is then critical [56,58]. However, the full penetrance associated with some VVCC may not be apparent until adulthood [55]. One may wonder whether the detection of patients who will not be symptomatic before adulthood is desirable. The frequency of variants and genotypes in the general population should be considered to evaluate the risk of a particular genotype to cause clinical CF; in other words, the penetrance of clinical CF in individuals carrying a given genotype. This was previously shown to be very low for R117H [59], an observation which led to its withdrawal from NBS variant panels [27,32], and, more recently, for the T5 variants and other variants commonly identified in inconclusive cases [60]. Such epidemiological data emphasize the importance of documenting the penetrance of variants before widely implementing EGA in core NBS programs. 

The increased detection of inconclusive cases when using EGA strategies should thus be anticipated with regard to the organization of CF centers, the costs of follow-up, and the additional investigations for variant interpretation. Moreover, additional studies on psychological consequences [61] of the long-term follow-up of these patients are needed before implementation of EGA as a global model.

##### Carrier Detection and Genetic Counseling 

Detection of healthy carriers, heterozygous for one CF-causing variant, is another unwanted result of NBS, though it has been considered a benefit that enables reproductive planning [62]. It will inevitably be increased by a more extensive DNA analysis in NBS strategy [10] and will have a cost for increased referrals in the families. Indeed, once a CF-causing variant is detected, a “cascade” genetic screening is recommended in relatives [63,64]. 

Here, again, as for inconclusive cases uncertainty about VUS and VVCC is puzzling and complicates genetic counseling. It is thus not only the higher number of healthy carriers of known CF-causing variants that healthcare professionals have to deal with, but the even higher expected number of carriers of VVCC and VUS. Updated information should be provided to families as knowledge evolves on variant pathogenicity, but such recalls could also generate anxiety. Population genetics data should not be neglected, as it has been shown that a number of variants classified as VUS or VVCC in CFTR2 actually have a frequency in the general population that argues against a severe impact, as for c.2620-26A>G, R74W, R117H, I556V, or D1270N [65]. The risk is to consider neonates as carriers of a CF-causing variant and to offer inappropriate genetic counseling and testing in the family, and eventually inappropriate prenatal diagnosis. In view of the expanding strategies that include EGA, guidelines should be issued on these matters to minimize as much as possible these byproducts. 

## 4. Conclusions and Perspectives 

Considerable expansion has been observed in the implementation of NBS programs for CF, with DNA analysis being part of an increasing number of strategies. While testing for *CFTR* gene variant panels as a second tier has been proven to improve the performance and reduce the false-positive rates, the value and place of *CFTR* EGA before the diagnostic step of sweat testing is still not obvious. The performance is indeed the result of all steps, starting with IRT cut-off values. Implementation of EGA in the NBS process, which has been greatly facilitated by advances in genomic sequencing technologies, is expected to achieve an optimal sensitivity in ethnically diverse populations and a faster genetic diagnosis in CF infants without the need of another sample. EGA has been implemented in three European countries and three US states only, with different mutational backgrounds and different protocols, so that comparison remains difficult. The major drawback of approaches that include EGA is the unwanted detection of neonates with an inconclusive diagnosis, with a CF:CFSPID ratio varying from one program to another, but which is as low as 1.2:1 in Poland [1] and 0.65:1 in California [7]. This challenging situation, leading to uncertainty for healthcare professionals and families, also generates costs that should be considered in the health economic evaluation of a program before its implementation. Based on data indicative of a low penetrance for a set of variants identified in inconclusive cases [59,60], it is likely that the detection of neonates carrying a true CF-causing variant in *trans* of such a variant would considerably increase in a program based on IRT/EGA, thus finally requiring a high number of sweat tests to be performed, not to mention other tests, follow-up visits, and parents’ resulting anxiety. Therefore, the significant risk reported for neonates with an inconclusive diagnosis to develop a CF disease [58] might be considerably lowered as many more cases are detected through EGA before sweat testing. By contrast, removal of R117H from the variant panel of the French NBS program led to dramatically improved performance, with the PPV increasing from 16% to 34% and the ratio CF:CFSPID from 6.3:1 to 9:1 [32]. 

Eventually, based on technological advances in genomic sequencing and the reduction of costs along with the steadily increasing implementation of NBS programs in multi-ethnic populations or populations where common variants are not included in current DNA panels, an optimal compromise to improve performance and minimize side effects would be to perform EGA with bioinformatics targeting a wide panel of fully penetrant CF-causing variants, including large deletions and deep-intronic variants, as recently implemented in New York State. EGA without applying any bioinformatic filter or without any additional step as IRT or PAP to limit the number of infants subjected to sweat testing might be detrimental, as long as the clinical impact and penetrance data associated with variants are not well documented. In view of this, the collection of data on the extended follow-up of neonates detected with an inconclusive diagnosis is of the utmost importance [58].

The question of the role of EGA in the context of NBS does not only focus on CF. Whole exome sequencing and whole genome sequencing are nowadays affordable due to a significant reduction of costs. The relevance of implementing extended NBS for numerous genetic diseases is currently debated [66,67,68], which again raises the unsolved question of the interpretation of rare variants all along the genome. In this respect, comprehensive information on genetic issues should be given to the parents prior to NBS. Expanded preconception carrier screening for recessive disorders is also under consideration in countries following an overall positive attitude of the general population [69], depending on ethnical and geographical origin, degree of consanguinity, and type of disorders. Preconception CF carrier screening has already been implemented in the US, Israël, and Northeast Italy [70]. Such a health public policy aims to detect most at-risk couples and this ineluctably impacts on the timing for a CF diagnosis and on the prevalence of CF births [71], which would then raise the question of the relevance of NBS. Reproductive attitudes would also inevitably change, as recently underlined [72]. Requests for prenatal diagnoses may increase, especially where non-invasive procedures have been recently available [73,74], but where a diagnosis of CF is made, with the growing availability of mutation-guided therapy the option of the continuation of pregnancy could be preferred over termination. 

In conclusion, designation of an optimal NBS program remains necessarily country-dependent. The introduction of EGA as a second tier without filtering of CF variants has major drawbacks that are yet unsolved and may negatively impact the organization of care and the cost-effectiveness of NBS for CF. Consideration of any change in strategy should be carefully evaluated, planned, and monitored. This reinforces the need to adopt a standardized approach to collect data to the benefice of performance comparison. Taking all of these elements into account will ensure the sustainability of implemented NBS programs.

## Figures and Tables

**Figure 1 IJNS-06-00023-f001:**
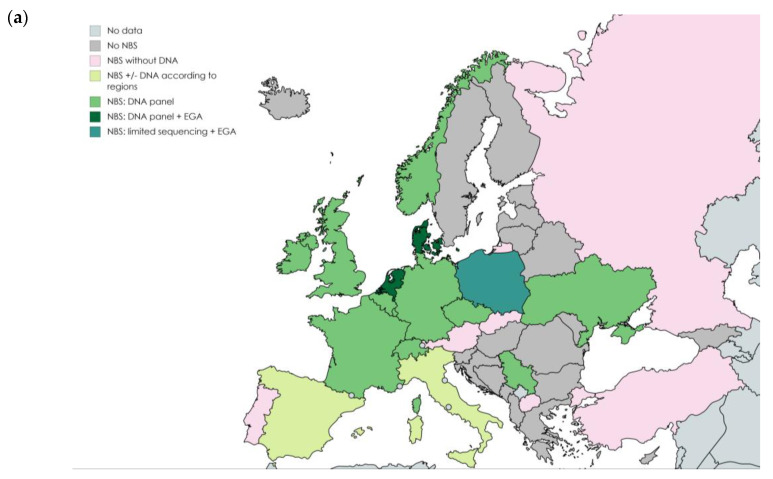
(**a**) In Europe, (**b**) in North America, (**c**) in South America, (**d**) in Australia and New Zealand. Newborn screening programs for cystic fibrosis (CF) according to the place of DNA analysis. NBS, newborn screening; EGA, extended *CFTR* genetic analysis.

**Table 1 IJNS-06-00023-t001:** Strategies and performance of newborn screening programs for CF.

Countries/States	1st Tier	2nd Tier	3rd Tier	4th Tier	Safety Net	IRT1 > Cut Off	Sensitivity (wo MI)	PPV CF	Ratio CF:CFSPID	Carrier Frequency
Brazil (Sao Paulo) [16]	IRT	IRT				0.73–1.67%	86–100%	3–19%	ND 1.1:1 (Turkey)	NA
Russia [22]										
Slovakia [1]										
Turkey [23]										
Spain (Andalusia) [24]										
Austria [25]		PAP			IRT	0.97%		23%	25:1 *	NA
Portugal [26]						0.70%	94.4%	41%	ND	NA
Germany [27,28]		PAP	DNA (31)		ST	0.73%	96%	20%	5:1	1/44
US (Colorado, Texas, Wyoming) [29]		IRT	DNA (41–48)		ST	2.10%	96%	20%	10.8:1	1/13
US (Wisconsin) [30]		DNA (25)				ND	95%	9%	5.2:1	1/9.5
Australia (Victoria) [18]		DNA (12)				ND	96%	18.3%	7.8:1	ND
New Zealand [19]		DNA (3)				ND	100%	23%	ND	ND
Italy (Tuscany) [31]		DNA (66)				0.85%	89.5%	19.4%	2.85:1	1/16
France [32]		DNA (29)			IRT	0.50%	95%	34%	9:1	1/16
Switzerland [33]		DNA (7)			IRT	0.78%	97%	36%	17:1 *	1/11 *
Czech Republic [1,34]		DNA (50)			IRT	0.90%	94%	15%	7.5:1 *	1/21
Norway [35]		DNA (72)	DNA (20)		ST	0.8%	95%	43%	1:1	1/10
UK [1,36]		DNA (4)	DNA (29 or 31)	IRT	IRT	0.57% *	96%	76%	10.5:1 *	1/28 *
Denmark [21]		DNA (1)	EGA		EGA	3.70%	92%	85%	7:1	1/20
California [7]		DNA (40)	EGA			1.6%	92%	34%	0.65:1	1/25
Netherlands [37]		PAP	DNA (35)	EGA	EGA	0.98%	90%	63%	4:1	1/28
Poland [1,38]		DNA(limited seq)	EGA			0.6% *	100%	26%	1.2:1 *	1/15 *

CFSPID, CF screen positive, inconclusive diagnosis; DNA, refers to either variant panel analysis, with the number of variants screened indicated in brackets, or to limited *CFTR* gene sequencing (limited seq) in Poland; EGA, extended *CFTR* genetic analysis; IRT, immunoreactive trypsinogen; NA, not applicable; ND, not documented; ST, sweat test; PAP, pancreatitis-associated peptide; PPV, positive predictive value; wo MI, without meconium ileus. * Figures taken from [1]. Since the sensitivity of the panels used were not always available, they were not indicated in the table.

**Table 2 IJNS-06-00023-t002:** Strengths and weaknesses of extended *CFTR* gene analysis in newborn screening programs for CF, as a second- or third-tier step (IRT/EGA or IRT/DNA panel/EGA), compared to strategies including tests for variant panels only (IRT/DNA panel).

Strengths	Weaknesses
**High PPV**	**but…**
Reduction of false positive results of NBS strategy with EGA in the third tier, hence: Reduction of unnecessary sweat tests with EGA in the third tier	Low CF:CFSPID ratio, expected to be further lowered if using EGA in the second tier because of: Numerous babies with two *CFTR* VUS or VVCC with EGA in the second tier, leading to unnecessary sweat tests, thus at the expense of PPV The important need for follow-up of infants with CRMS/CFSPID
**High Sensitivity**	**but…**
Highest expected sensitivity with EGA in the second tier Best equity between populations, adapted for minorities carrying rare variants Unmasking new couples at risk of having a CF child (carrier detection)	Misdiagnosis of CF or CRMS/CFSPID if referring to CFTR2 only Increased number of carriers detected, including carriers of VUS and VVCC, especially with EGA in the second tier
**Technically feasible**	**but… requires optimal healthcare system organization**
NGS already in routine in laboratories for several years Coverage of all known and unknown variants	NBS sequencing platform development or reorganization, especially if EGA in the second tier Questionable cost-effectiveness of the NBS program because of its impacts beyond Technical limitations (homopolymers, large deletions, deep intronic variants) depending on the technique and the design
**Increased knowledge on *CFTR* variants**	**but… raises questions on variant interpretation**
Database enrichment and genotype–phenotype studies Genetic counseling for future cases/pregnancies	Increased number of VUS Need for additional in vitro/ex vivo experiments and clinical examinations for variant interpretation Disease penetrance unknown for many variants
**More precise medical care**	**but… raises ethical questions**
Detection of the two disease-causing variants in a single step Earlier access to treatment	Possible delay at the first medical visit according to ECFS recommendations [14] Psychological impact of long-term follow up of infants with CRMS/CFSPID Detection of patients who may not be symptomatic before adulthood and are at an unknown risk of developing a CFTR-related disorder

CFSPID, CF screen positive, inconclusive diagnosis; CRMS, CFTR-related metabolic syndrome; EGA, extended *CFTR* genetic analysis; IRT, immunoreactive trypsinogen; NBS, newborn screening; NGS, Next Generation Sequencing; VVCC, variant of varying clinical consequence; VUS, variant of uncertain or unknown significance.

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
