# Peer review of "The Role of Extended CFTR Gene Sequencing in Newborn Screening for Cystic Fibrosis"

_2409-515X, 2020, doi:10.3390/ijns6010023_

Round 1

Reviewer 1 Report

This is a well-written review. Overall the review is well structured, easy to read and consists of all important information needed.

I have a few minor comments:

Table 1 - column "safety net": Does "TS" mean sweat test (ST)?

Line 171 - Best equity of CF screening between populations: PAP-based protocols are also thought to be beneficial for multiethnic populations. What are the advantages/disadvantages of EGA when compared to PAP when equity in multiethnic populations is desired? Please add also a shor statement in the text of the manuscript!

Line 228 - "Shorts insertions": Did you mean "Short insertions"?

Author Response

We thank the reviewers for their helpful comments. Below are in blue the point-by-point replies to the reviewer’s comments (restated in black). The changes made are seen as "track changes" in the revised manuscript.

Reviewer 1

Table 1 - column "safety net": Does "TS" mean sweat test (ST)?

R: We have changed TS for ST.

Line 171 - Best equity of CF screening between populations: PAP-based protocols are also thought to be beneficial for multiethnic populations. What are the advantages/disadvantages of EGA when compared to PAP when equity in multiethnic populations is desired? Please add also a shor statement in the text of the manuscript!

R: We acknowledge that protocols relying on biochemical tests theoretically avoid discrimination between populations. The point in this review was to discuss the place of EGA among strategies using DNA analysis, underlining that introduction of EGA in NBS protocols is intended to achieve a better sensitivity in multiethnic populations, as compared to variant panels. The comparison between IRT/PAP and IRT/DNA/EGA or IRT/PAP/DNA/EGA strategies does not seem at the advantage of the IRT/PAP protocol in terms of sensitivity and PPV (Dankert-Roelse, J Cyst Fibros 2019). PAP has been introduced in the NL protocol (IRT/PAP/DNA/EGA) to the benefit of a lower rate of CFSPID and carriers. PAP-based protocols have been shown to lead to a substantial rate of false negative cases, regardless of the ethnic origin of these missed cases. This could be modulated by lowering the PAP cutoff, but at the expense of PPV. Moreover, to our knowledge, no data is available on PAP variations in multiethnic populations. PAP has not been introduced in NBS programmes outside Europe, so that “its benefits and harms in a pan-ethnic community have not been clarified” (to cite Ross, J Pediatr 2008).

For these reasons, it does not seem relevant to add a sentence comparing IRT/DNA±EGA with IRT/PAP protocols.

Line 228 - "Shorts insertions": Did you mean "Short insertions"?

R: We have changed « shorts insertions » for « short insertions ».

Reviewer 2 Report

This interesting article focuses on the impact of extended CFTR gene sequencing in Newborn screening for cystic fibrosis, both giving some answers and raising new questions. This paper is to be read carefully before implementing more widely extended gene analysis in Newborn screening for cystic fibrosis and other genetic diseases.

Line 29: as far as IRT is concerned, the first NBS programmes were implemented 40 years ago, rather than 50.

Line 78: The maps of Australia and New-Zealand are missing

Table 1: The DNA panel used in France is made of 29 variants since the R117H was suppressed in 2015. In the caption of Table 1, "neg" & "negative sweat test" are not necessary. ST is sufficient and should replace TS in the Table. 

Line 110: what is an independent sample? Is it a second punch of dried blood which is retested to verify an elevated IRT? If it is so, France does the same.

Line 115: same question.

Line 163 to 310: This long discussion with two parts (values and challenges) of the EGA introduction in NBS for CF is largely a copy of table 2. So, I suggest it could be shortened.

Author Response

We thank the reviewers for their helpful comments. Below are in blue the point-by-point replies to the reviewer’s comments (restated in black). The changes made are seen as "track changes" in the revised manuscript.

Reviewer 2

Line 29: as far as IRT is concerned, the first NBS programmes were implemented 40 years ago, rather than 50.

R: This has been changed.

Line 78: The maps of Australia and New-Zealand are missing

R: The maps of Australia and New Zealand are present in manuscript. They are on another sheet just before the caption of Figure 1.

Table 1: The DNA panel used in France is made of 29 variants since the R117H was suppressed in 2015.

R: This has been changed.

 In the caption of Table 1, "neg" & "negative sweat test" are not necessary. ST is sufficient and should replace TS in the Table. 

R: This has been changed in Table 1 and its legend.

Line 110: what is an independent sample? Is it a second punch of dried blood which is retested to verify an elevated IRT? If it is so, France does the same.

R: The 2nd IRT measurement is made on a new blood sample, at 2 weeks of age. To make it clear, « independent blood sample » has been changed for « new blood sample ».

Line 115: same question.

R: Again, to make it clear, « an independent sample » has been changed for « a new sample ».

Line 163 to 310: This long discussion with two parts (values and challenges) of the EGA introduction in NBS for CF is largely a copy of table 2. So, I suggest it could be shortened.

R: The discussion was indeed long and has been shortened. If Table 2 summarizes strengths and weaknesses, details are necessary to explain in particular technical and ethical barriers.

Sentences covering the following lines have been deleted:

  • 176-181
  • 187-190
  • 203-207
  • 210-213

Lines 226-238 have been modified so that the paragraph is shortened and the order of references 50 to 52 has been changed.